# Effect of Xylo-Oligosaccharides Supplementation by Drinking Water on the Bone Properties and Related Calcium Transporters in Growing Mice

**DOI:** 10.3390/nu12113542

**Published:** 2020-11-19

**Authors:** Hang Gao, Zhenlei Zhou

**Affiliations:** College of Veterinary Medicine, Nanjing Agricultural University, Nanjing 210095, China; 2019207053@njau.edu.cn

**Keywords:** xylo-oligosaccharides, bone properties, calcium transporters, growing mice

## Abstract

Xylo-oligosaccharides (XOS), non-digestible oligosaccharides, have the potential to regulate intestinal microorganisms, and thus, improve host health, but little evidence exists for the prebiotic effects on bone health. This study evaluates the dose-response effect of XOS supplementation on bone properties, the morphology of the intestine, cecum pH, and cecum wall weight, as well as the related calcium transporters. Ninety-six 28-day-old male mice were randomized into one of four groups, fed the same commercial diet, and given different types of deionized water containing 0, 1, 2, or 4% XOS by concentration for 30 days. Eight mice were randomly selected to accomplish particular tasks every 10 days. No significant differences in serum Ca and P levels and growth performance were observed among the four studied groups. XOS intervention significantly decreased cecum pH and increased cecum wall weight in a dose-dependent manner. At the late growth stage, compared with 0% XOS, the bone mineral density (BMD) and bone-breaking strength in 4% XOS were significantly higher. The bone crystallinity with 4% XOS, measured by Raman spectrum, was significantly enhanced compared to that with 0% XOS during later growth. The villus height and villus height to crypt depth (VH:CD) were enhanced with an increase of XOS concentration during the later stage of growth. The expression of transient receptor potential vanillin receptor 6 (TRPV6) and Na^+^/Ca^2+^ exchanger 1 (NCX1) in the duodenum were enhanced by XOS supplementation. XOS exerted a positive influence on bone properties by decreasing the cecum pH, increasing the cecum wall and villus structure, and upregulating the expression of related calcium transporters.

## 1. Introduction

One approach to reducing the risk of osteoporosis later in life for children and adolescents is to ensure the development of ideal peak bone mass and bone strength during growth [1]. Any enhancement in calcium (Ca) absorption during childhood and adolescence may ultimately increase bone retention, which will help achieve optimal peak bone mass. The supplementation of dietary calcium from milk and dairy products is the main source of Ca. Ca supplementation in tablet form is also one of the basic measures. At present, the public prefers to focus on the functional ingredients that improve Ca absorption [2]. Functional oligosaccharides, also known as non-digestible oligosaccharides (NDOs), are currently considered the most promising prebiotics for improving bone health [3,4,5]. Examples of these non-digestible carbohydrates include galactooligosaccharide (GOS), fructooligosaccharide (FOS), human milk oligosaccharides (HMO), isomaltose oligosaccharides, xylo-oligosaccharides (XOS), and inulin. The non-digestibility of NDOs in the gastrointestinal tract is due to resistance to gastric acid, mammalian hydrolases, and gastrointestinal absorption. After undergoing fermentation by the intestinal microbiota, NDOs selectively stimulate either the growth or the activity of gut microbiota with demonstrable benefits for host health [6]. The main nutritional role of NDOs is to produce a direct physiological effect on the host, including anti-inflammation, antioxidation, mineral metabolism, lipid metabolism, and immune modulation [3,4,7,8,9].

XOS is a kind of emergent prebiotic composed of 2–7 xylose molecules with β-1,4 glycoside bond [10], which are naturally found in fruits, vegetables, milk, and honey [11]. XOS supplementation has shown significant prebiotic activities in humans, broilers, laying hens, and weaned piglets by modulating the gut microbiota [12,13,14]. Presently, XOS is widely used in health care, dairy beverages, and the food and feed industries. However, people currently attach great importance to XOS, mainly because of its unique effects on enhancing animal disease resistance, stimulating the immune system of animal bodies, and improving a number of chronic, inflammatory conditions. Compared with other NDOs, such as GOS, FOS, and inulin, few data about XOS are available that support the evidence of mineral metabolism and bone health in an animal model [15]. One of these studies reported that the probiotics *L. paracasei* HII01, XOS, or both together as a biotic therapy produced similar potential enhancement of bone microarchitecture in rats fed a high-fat diet (HFD), possibly by mitigating osteoclast-mediated bone resorption and promoting osteoblast-induced bone formation [16]. In addition, previous research into dietary XOS on bone mineral crystallinity in swine femurs suggested that XOS enhances the crystallinity of bone and that the effects of XOS on femurs may be more obvious during periods of early growth, while this influence was not observed in bones sampled during later growth [17]. These studies suggest that XOS may be involved in the absorption of minerals and improving bone health. Currently, there are two widely accepted mechanisms used to explain NDO-bone interactions: (1) The production of short-chain fatty acids (SCFAs) in the lower gut by intestinal microorganism fermentation and (2) a change in the microbial community composition. It is also worth considering the changes in tissue morphology and mineral transporter proteins. Moreover, few reports about inulin, lactose, and other prebiotic NDOs have indicated their molecular mechanisms of mineral absorption [18,19,20].

The majority of the literature supporting the effects of prebiotics on bone comes from animal research. Hence, rodent models are imperative for understanding the direct contributions of the XOS-induced changes to the prebiotic effects at the level of bone health. The aim of this study was to determine, in growing mice, the effects of XOS provided by water on bone-relevant parameters and the expression of related calcium transporters using methods of biomechanics, Raman spectroscopy, and immunohistochemistry. We hypothesize, through a rodent modeling analysis, that XOS enhances bone properties and can be used as an adjuvant therapy to achieve peak bone mass in the growing period.

## 2. Materials and Methods

### 2.1. Animals, Diets, and Experimental Design

Ninety-six 21-day-old weanling male Institute Cancer Research (ICR) mice (20–22 g) were purchased from Qinglongshan Animal Breeding Farm in Nanjing. They were housed in individual stainless-steel cages under standardized environmental conditions (22 ± 1 °C, the relative humidity of 50%, on a 12 h light/dark cycle) in the Nanjing Agricultural University animal facility. After a 7days adaptation period, ninety-six 28-day-old mice were randomly assigned to the four studied groups to receive one of the four types of experimental drinking water until 58 days of age. All groups were fed the same commercial cereal-based diet (Qinglongshan Feed Production Co., Ltd., Nanjing, China; containing 18% crude protein, 5% crude fiber, 4% crude fat, 8% crude ash, 1.0% Ca, 0.6% P, and 5.3% methionine and cysteine), and XOS (95% XOS, Longlive Bio-Technology Co., Ltd., Dezhou, China) was given at concentrations of 0, 2.1, 4.2, and 8.4 g for every 200 mL of deionized water (1.0% XOS group, control group; 2.1% XOS group; 3.2% XOS group, and 4.4% XOS group). The whole experiment lasted 30 days, and during this period, eight mice from each group were randomly selected to accomplish particular tasks every 10 days (*n* = 8). The bodyweight and food intake were recorded once every 10 days. The experimental drinking water was recorded every day. Throughout the entire experimental period, all mice had free access to food and experimental drinking water. Food efficiency ratios (FERs) for the experimental period were calculated by the bodyweight gain divided by food intake, and signs of health were observed daily. However, no adverse events or mortality occurred during the trial.

All animal maintenance, handling, and procedures were performed in accordance with the National Institutes of Health guidelines for the care and use of laboratory animals and were approved by the ethical committee at Nanjing Agricultural University (Approved number NJAU-VM-2018009).

### 2.2. Sample Collection

At 38, 48, and 58 days of age, eight mice without food and water for 8hours from each group were randomly selected and euthanized with a CO_2_ overdose to accomplish many tasks. Fasting blood samples were collected from the retro-orbital venous plexus before sacrifice, and the serum samples were stored at −20 °C until the analyses were performed. Cecum pH was measured directly by a pH meter (SX-620, Sanxin, Shanghai), and the cecum contents were removed. The cecum wall was then flushed with 0.9% saline, blotted dry with filter paper, and weighed. The duodenum was separated and stored in 4% formaldehyde for histology and semi-quantitative immunohistochemistry. Femurs were excised and cleaned of their muscle tissue and stored at −20 °C for later bone properties analyses.

### 2.3. Determination of Serum Calcium and Phosphorus

The calcium and phosphorus (P) concentration in serum was evaluated through habitual methods using an automated analyzer (Bs-300, Mindray Biomedical Electronics Co., Ltd., Shenzhen, China).

### 2.4. Determination of Femur Biological Parameters

The left femurs were removed from −20 °C and thawed overnight at 4 °C. Bone growth parameters included length and diameter (middle axis of bone) were measured in millimeters by vernier caliper, and bone weight was measured in grams by the precision electronic scale.

The dried left femur samples were placed in the center of the stage, bone mineral density (BMD) and bone mineral content (BMC) were measured by dual-energy X-ray absorptiometry (DEXA; InAlyzer, Medikors, Korea), after the calibration of the instrument. All femur samples were scanned using identical scan parameters (55 kVp/1.25 mA~80 kVp/1.0 mA) and placed in the same position. After scanning, a digital image analysis system (InAlyzer1.0, Korea) was used to analyze the achieved image. Then, different subareas were analyzed on the image of the femur on the screen using a region of interest (ROI) for each segment. Finally, the pictures and data were saved.

The widths of Raman peaks are often used to evaluate bone mineral crystallinity. Raman microprobe spectroscopy (ThemoFisher DXR532, Madison, WI, USA) was performed on the middle part of the cross-section (sawn across the mid-shaft) of the left femur. Some relevant parameter settings are as follows: The spectral region, 400–1800/cm; laser, 532 nm; laser power, 10 mW. Raman analysis of each sample was performed three times (three different spots). The peak positions were calibrated with a silicon wafer (520.5/cm). All Raman spectra were acquired through the typical acquisition of the 4 × 4 s per analysis spot using an LMPlan 20× objective (Olympus, Inc., Tokyo, Japan). The data were analyzed with the Labspec 5 software.

Lastly, the breaking strength at the midpoint of each left femur was determined by a three-point bending test using a material testing machine (LR10K PLUSLloyd Instruments Ltd., Bognor Regis, UK). The load was applied vertically to the long axis at the mid-length region of the femur (displacement rate of 15 mm/min, preload of 5 N). The distance between the supporting points was 8 mm. Once the femur sample broke, the machine was stopped, and then a compression curve appeared on the screen. The compression curve was analyzed and processed with the NEXYGEN PLUS software. The highest value of the curve was the bone-breaking strength (N) of the test sample. The data were then recorded.

### 2.5. Gut Morphology and Immunohistochemistry Analysis

After euthanization, a 2 cm segment was removed from the duodenum, washed in a physiological saline solution, and fixed in 4% formaldehyde for 24 h. Then, the samples were dehydrated through consecutive embedding in graded ethanol solutions. Thereafter, the fixed samples were embedded in paraffin. Transverse and longitudinal sections 5 μm in thickness were prepared using a microtome stained with hematoxyline-eosin. Finally, the prepared tissue slices were used for morphological observations. Villus height and crypt depth were examined under a light microscope (OLYMPUS DP71, BX50F-3, Olympus Optical Co. Ltd., Tokyo, Japan). The examinations were made under a 40× lens, all measurements were obtained using 15 villi per slide, and the mean values of 2 slides per pen were used for statistical analysis.

Five-micron-thick tissue sections from another 2 cm duodenum segment were deparaffinized in xylene and rehydrated in a descending series of ethanol in distilled water. The tissue sections were treated with 3% hydrogen peroxide methanol solution for 15 min, followed by three 5 min rinses with phosphate-buffered saline. Then, the sections were blocked with 10% normal goat serum for 40 min at room temperature. The sections were incubated overnight at 4 °C with either anti-rat matrix transient receptor potential vanillin receptor 6 (TRPV6) (1:500; 15506R from Bioss, Beijing, China) or anti-rat matrix Na^+^/Ca^2+^ exchanger 1 (NCX1) (1:1000; ab2869 from Abcam, Cambridge, UK). The color was developed by incubation in DAB (ZSGB-BIO, Beijing, China). The sections were counter-stained with hematoxylin. All sections were semi-quantitatively analyzed using Image-Pro Plus. The average optical density (AOD) was determined by the ratio of integrated optical density (IOD) to the area (region of interest) in three pictures for each sample.

### 2.6. Statistical Analysis

Data were analyzed using SPSS version 20.0 for Windows (IBM Japan, Ltd., Tokyo, Japan). All data were presented as the mean and standard errors (SE). After verification of a normal distribution, a one-way analysis of variance (ANOVA) followed by the Tukey multiple-comparisons procedure was used to compare means of the four groups. *p* values less than 0.05 were considered significant.

## 3. Result

### 3.1. Weight Gain, Food and Water Intake, Food Efficiency Ratio

There were no significant differences (*p* > 0.05), due to XOS, in the average weight gained or the average final weight among the groups for all mice combined (Table 1). Moreover, no significant changes (*p* > 0.05) in food and water intake or FERs were observed.

### 3.2. Cecum pH and Cecum Wall Weight

A significant (*p* = 0.005) and dose-dependent decrease in cecum pH were observed (Table 2). This decrease was accompanied by a dose-dependent increase in cecum wall weight, but only a significant difference (*p* = 0.01) was observed in 4% XOS compared to group 1 without XOS.

### 3.3. Serum Ca and P Concentration

No significant (*p* > 0.05) differences were found at 58-d between all groups when the serum Ca and P concentrations were analyzed (Table 2).

### 3.4. Bone Analysis

No significant differences (*p* > 0.05) among groups in the length, weight, or diameter of the middle axis of the femur were observed in the different stages of growth (Table 3). Though the BMD of the whole and distal femur determined via DEXA in all XOS groups showed a dose-dependent increase in the three stages of growth, a significant difference (*p* = 0.04) was seen only in the distal femur BMD between 0% XOS and 4% XOS in the period of later growth. Supplementation of different doses of XOS had no significant effect on femoral BMC in mice during different growth periods (Appendix A). Meanwhile, the bone breaking strength of the 4% XOS group was significantly higher (*p* = 0.008) when compared to the 0% XOS group in the late growth stage.

Raman spectroscopy was performed on the femurs of all mice. The 960/cm peak is commonly regarded as a measure of bone phosphate content [21,22]. The full width at half maximum (FWHM) of the 960/cm peak indicates the degree of atomic order in the crystallites, with wider peaks reflecting more atomically disordered materials [23,24]. The 960/cm FWHM of group 4 at 58-d was significantly (*p* = 0.007) lower than that of group 1 without XOS, and no significant difference was observed in the other groups (*p* > 0.05). However, at 38 and 48 days, the 960/cm FWHM in all groups showed no significant (*p* > 0.05) differences (Table 3).

### 3.5. Gut Morphological Evaluation and Immunohistochemistry Analysis

Duodenum tissue slices were used for morphological observations. As the drinking XOS concentration increased, the villus height of the duodenum showed a linear increase (Table 4). However, only 4% XOS reached statistical significance (*p* < 0.001, *p* = 0.001, and *p* = 0.008, respectively) compared with 0%, 1%, and 2% XOS. Meanwhile, a dose-dependent decrease in crypt depth was observed (Table 4), and a significant increase was only observed between the 0% XOS and 4% XOS groups (*p* = 0.006). Naturally, the change in VH:CD was consistent with the intestinal villus height and crypt depth. The use of 4% XOS significantly increased the VH:CD compared to 0% XOS and 1% XOS (*p* = 0.001 and *p* = 0.004, respectively).

Immunohistochemistry assays and semi-quantitative analyses were performed to evaluate the protein expression of TRPV6 and NCX1 in the duodenum of all 58-d mice (Figure 1). Positive staining with NCX1 was observed in all groups. Compared with the control group without XOS, the 1%, 2%, and 4% XOS groups all had upregulated expression of NCX1, but a significant increase was only observed between the control and 4% XOS group (*p* = 0.01). Meanwhile, the expression of TRPV6 in the 1%, 2%, and 4% XOS group mice was significantly higher than the expression in the control group (*p* = 0.02, *p* = 0.03, and *p* = 0.04, respectively), while there was no significant difference found among the three groups: 1%, 2%, and 4% XOS (*p* > 0.05).

## 4. Discussion

Although the effects of several types of NDOs, such as GOS and FOS, on mineral absorption, intestinal parameters, and bone health have been reported [25,26,27,28], little is known about the effect of XOS supplementation on the potential role of bone and related calcium transporters. The rodent model used to conduct the present study showed that XOS has a positive effect on bone parameters, especially during later growth. Moreover, the greatest benefits to bone properties from XOS consumption were observed at levels of 4%.

In this study, no significant changes in growth performance-related parameters were seen in the four dose-gradient groups. Indeed, irrespective of the XOS concentration in the drinking water, the food consumption, efficiency, and bodyweight gains were similar in all groups, due to eating the same isocaloric diet. Moreover, XOS supplementation did not affect the water intake of mice, and there were no significant differences among groups. The effect of XOS supplementation on growth performance was not the focus of this trial, and the results were as varied as most other NDOs indicate. In addition, XOS also had no effect on serum calcium and phosphorus concentrations because calcium and phosphorus metabolism are regulated and controlled by multiple hormones in the body.

Unsurprisingly, XOS supplementation exerted significant effects on both cecum pH and cecum wall weight in a dose-dependent manner, similar to other NDOs [29,30]. The decrease in cecum pH, possibly due to the production of SCFAs, created a more acidic environment for the intestinal cavity [31,32]. This acidic environment was more conducive to improving the solubility of minerals, thereby increasing the absorption of minerals. In addition, the weight gain of the cecum wall may indirectly result from the presence of SCFAs, since butyrate is reported to be one of the primary energy sources for intestinal epithelial cells [18,33,34]. The production of SCFA and its nutritional effect on cecum cell proliferation were not evaluated in the present report. However, intestinal mucosal morphometry was evaluated in this study, including intestinal villus height, crypt depth, and VH:CD. XOS supplementation significantly increased the height of the duodenal villus, decreased the crypt depth, and naturally also increased the VH:CD, which indicates that the duodenal absorption area increased. This increase in the absorption surface area contributed to the absorption of minerals in the intestine. Similar to our results, some reports indicated that the duodenal villus height and crypt depth are also affected by XOS supplementation in chickens [13,35]. Because of the scarcity of available reports on the effects of XOS on villus height and crypt depth in rodent models, a comparison was made with other studies that used similar functional oligosaccharides. Although it was reported that NDOs undergo fermentation via intestinal microbiota in the lower intestine, Darío Pérez-Conesa et al. [36] observed that no test diet significantly modified the crypt depth or cell density in the cecum via GOS or synbiotics. Due to the dominance of the small intestine in digestion and absorption, alongside the non-influence of GOS on cecal morphometry, the focus of this trial was shifted to the duodenum. The duodenum is the main site for the transcellular transport of Ca^2+^ [37]. The expression of related calcium transporters in the duodenum was analyzed by immunohistochemistry in this test. Compared with 0% XOS, 4% XOS supplementation significantly enhanced the expression of NCX1, while 1% and 2% versus 0% XOS did not achieve a statistical difference. By contrast, the expression of TRPV6 in 1%, 2%, and 4% XOS was greater than that of in 0% XOS. Hence, XOS likely upregulated the calcium transporters, such as NCX1 and TRPV6, in the active transcellular pathway of Ca^2+^, thereby promoting calcium absorption.

The observed effects on cecal wall weight and pH, intestinal mucosal morphology, and calcium transporters indirectly suggest that all the above mechanisms were likely involved in the improvement of bone properties found in the present study. The positive effects of XOS supplementation on mineral absorption possibly contributed to an increase in Ca bioavailability. The increase in Ca absorption efficiency provided an extra supply of minerals, the main inorganic components of bone, inducing a beneficial increase in almost all the studied bone parameters associated with bone quality. Human bone mineral content is closely related to bone strength and internal environmental stability, so BMD is considered an important index to evaluate human health. The accurate content of bone minerals (mainly calcium) obtained directly plays an important role in judging and studying skeletal physiology, pathology, and human aging, as well as in diagnosing various diseases of the whole body [38]. Although the femur BMD increased in a dose-dependent manner with XOS intervention, no statistically significant difference was observed in any groups. Indeed, the specific bone sites that benefitted most had higher amounts of trabecular bone, which is metabolically more active than cortical bone (that is, the distal femur). Of course, the BMD of the distal femur by DEXA was positively influenced by 4% XOS in this study. In addition, at the midshaft femur, the breaking strength was improved by XOS. Similarly, the greatest benefit was seen with 4% XOS treatment. The strength of the newly formed bone depends on the architectural disposition of bone material, as well as other factors that are unrelated to mineralization, such as crystal arrangement and size [39]. Raman spectroscopy can be used to assess the chemical properties of minerals and organic molecules, such as measuring mineral crystallinity, and 960/cm FWHM are often used to investigate mineral crystallinity, where the wider the peak is, the more atomically disordered the material is [40]. Similar to the breaking strength, femoral mineral crystallinity benefitted most in 4% XOS group. Moreover, it was not unexpected that the studied bone parameters enhanced significantly in the period of later growth because of the longer duration of administration [17].

In conclusion, XOS supplemental exerted a positive effect on bone properties, possibly induced by cecal wall weight and pH, intestinal mucosal morphology, and related calcium transporters. Hence, an appropriate dose of XOS can be regarded as a promising adjunct strategy for teenagers to achieve peak bone mass.

## Figures and Tables

**Figure 1 nutrients-12-03542-f001:**
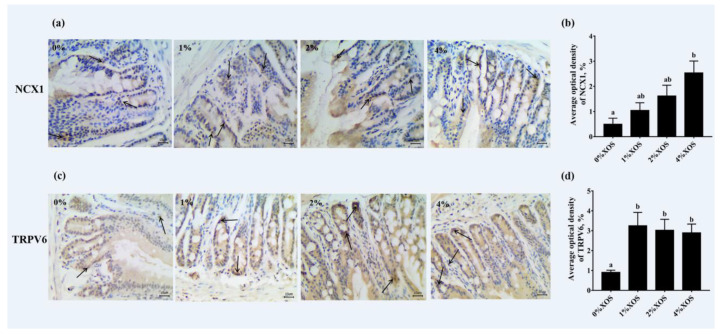
Effects of xylo-oligosaccharides (XOS) supplementation on the Na^+^/Ca^2+^ exchanger 1 (NCX1) (**a**,**b**) and transient receptor potential vanillin receptor 6 (TRPV6) (**c**,**d**) expression in the duodenum of the four groups. Scale bar: 10 µm. Values are means (n 8) with their standard errors represented by vertical bars. ^a,b^ Mean values with unlike letters were significantly different (*p* < 0.05; Tukey post hoc test).

**Table 1 nutrients-12-03542-t001:** Weight Gain, Food and Water Intake, and FER of all mice fed different levels of XOS (Mean values and standard errors; *n* = 8 in each group).

Variables	0% XOS		1% XOS		2% XOS		4% XOS	
	Mean	SE	Mean	SE	Mean	SE	Mean	SE
FI (g/day)	6.96	0.05	6.93	0.07	6.91	0.05	6.93	0.10
WI (mL/day)	5.96	0.07	5.98	0.14	5.94	0.16	5.85	0.12
IBW (g)	21.38	0.08	21.25	0.10	21.99	0.13	21.44	0.13
FBW (g)	29.04	0.68	28.99	0.71	29.65	0.67	29.13	0.60
BG (g)	7.66	0.24	7.74	0.16	7.66	0.20	7.69	0.24
FERs (%)	3.67	0.03	3.72	0.02	3.70	0.01	3.70	0.01

SE, standard error; FI, food intake; WI, water intake; IBW, initial bodyweight; FBW, final bodyweight; BG, bodyweight gain; FERs, food efficiency ratios. no superscript letter = no significance.

**Table 2 nutrients-12-03542-t002:** Cecum pH, Cecum Wall Weight, and Serum Ca and P Concentrations of 58-d mice fed different levels of XOS (Mean values and standard errors; *n* = 8 in each group).

Variables	0% XOS		1% XOS		2% XOS		4% XOS	
	Mean	SE	Mean	SE	Mean	SE	Mean	SE
Cecum pH	7.29 ^a^	0.01	7.11 ^b^	0.02	6.95 ^c^	0.02	6.82 ^d^	0.03
Cecum Wall Weight (g)	0.21 ^a^	0.01	0.22 ^ab^	0.01	0.23 ^ab^	0.01	0.24 ^b^	0.00
Ca (mmol/L)	2.17	0.02	2.19	0.02	2.11	0.01	2.11	0.02
P (mmol/L)	2.23	0.02	2.21	0.02	2.22	0.02	2.21	0.02

SE, standard error; ^a, b, c, d^ Mean values with unlike superscript letters within the same row are significantly different (*p* < 0.05; Tukey post hoc test).

**Table 3 nutrients-12-03542-t003:** Femoral parameters, mineral density, breaking strength, and FWHM of 960/cm peak of 38, 48, and 58-d mice fed different levels of XOS (Mean values and standard errors; *n* = 8 in each group).

Variables	Age (day)	0% XOS		1% XOS		2% XOS		4% XOS	
		Mean	SE	Mean	SE	Mean	SE	Mean	SE
Bone length (mm)	38	13.89	0.14	13.83	0.12	13.89	0.12	13.74	0.14
	48	14.73	0.13	14.57	0.13	14.70	0.15	14.74	0.16
	58	14.80	0.11	15.04	0.12	15.05	0.35	15.13	0.18
Bone weight (g)	38	0.54	0.01	0.53	0.00	0.55	0.00	0.60	0.01
	48	0.64	0.01	0.60	0.02	0.57	0.00	0.60	0.01
	58	0.61	0.02	0.63	0.00	0.61	0.00	0.58	0.01
Bone diameter (mm)	38	1.23	0.04	1.26	0.05	1.30	0.06	1.42	0.03
	48	1.36	0.07	1.42	0.04	1.41	0.05	1.47	0.02
	58	1.42	0.04	1.43	0.03	1.44	0.03	1.47	0.02
Bone breaking strength (N)	38	18.87	0.59	18.43	1.00	18.97	1.12	18.94	0.88
	48	20.32	0.64	19.89	1.09	21.35	0.64	22.70	1.23
	58	20.71 ^a^	0.28	21.45 ^ab^	0.61	22.04 ^ab^	0.71	23.29 ^b^	0.40
Femur BMD (mg/cm^2^)	38	92.65	3.14	90.77	2.11	95.55	3.90	95.00	3.54
	48	102.08	2.45	105.45	3.38	106.67	3.13	111.88	3.47
	58	110.52	3.50	112.14	4.01	113.91	2.17	119.55	1.25
Distal femur BMD (mg/cm^2^)	38	104.77	2.31	108.36	3.94	107.05	3.24	110.97	6.47
	48	122.27	4.38	118.68	3.97	122.97	4.40	134.68	3.24
	58	123.85 ^a^	2.17	132.03 ^ab^	6.24	133.43 ^ab^	4.83	145.10 ^b^	5.83
FWHM	38	17.52	0.37	17.63	0.42	16.63	0.20	16.43	0.19
	48	15.88	0.39	15.92	0.47	15.36	0.04	15.82	0.36
	58	16.74 ^a^	0.15	16.12 ^ab^	0.23	16.06 ^ab^	0.30	15.39 ^b^	0.17

SE, standard error; BMD, bone mineral density, FWHM, full width at half maximum ^a, b^ Mean values with unlike superscript letters within the same row are significantly different (*p* < 0.05; Tukey post hoc test); no letter = no significant.

**Table 4 nutrients-12-03542-t004:** Duodenum morphological measurement of 58-d mice fed different levels of XOS (Mean values and standard errors; *n* = 8 in each group).

Variables	0% XOS		1% XOS		2% XOS		4% XOS	
	Mean	SE	Mean	SE	Mean	SE	Mean	SE
Intestinal villus height	568.89 ^a^	17.85	587.21 ^a^	18.87	612.13 ^a^	25.20	699.23 ^b^	24.88
Crypt depth	90.80 ^a^	5.74	82.70 ^ab^	4.11	77.91 ^ab^	4.14	71.41 ^b^	2.51
VH:CD	6.58 ^a^	0.32	7.19 ^a^	0.31	7.98 ^ab^	0.38	9.28 ^b^	0.49

SE, standard error; VH:CD, villus height: crypt depth. ^a,b^ Mean values with unlike superscript letters within the same row are significantly different (*p* < 0.05; Tukey post hoc test).

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
