# Peer review of "Effect of Xylo-Oligosaccharides Supplementation by Drinking Water on the Bone Properties and Related Calcium Transporters in Growing Mice"

_nutrients, 2020, doi:10.3390/nu12113542_

Round 1

Reviewer 1 Report

Nice study design and results. Below are my comments which are intended to help improve the manuscript. Best regards.

Title:

Consider changing title to "Effects of drinking water supplementation of xylo-oligosaccharides on bone properties calcium transporters in growing mice.

Abstract:

Define ICP in 1st time use. Spell out and put ICP in parenthesis.

Words count in abstract can be reduced by converting to a non-structured abstract

Introduction:

Line 40:  How about adding human milk oligosaccharides (HMO) as a type of NDOs?

Line 72: Provide some comments on the rationale for using drinking water for supplementing XOS instead of a direct supplement of the commercial diet. How soluble is the XOS in the drinking water?

Methods:

Line 115: This sentence needs to be revised to clarify that length and diameter are in mm and weights are in grams.

Line 117: Modify this statement to read - "The dried left femur samples were placed in the center of the stage, and bone mineral density (BMD) was measured by dual energy X-ray absorptiometry (DEXA; InAlyzer, Medikors, South Korea), after the calibration of the instrument."

Line 123: Did you collect bone mineralization (BMC) alongside BMD? If yes, will you be willing to report as supplemental data? In addition, did you adjust BMD and BMC for weight and length because of autocorrelation of BMC and BMD to them?

Statistical Analysis:

Did you consider the use of repeated measures analysis in a 4 treatment groups by 4 time points?

Results:

In the Tables and the Figure, insert “=” between “n” and values.

Abbreviations:

I am not sure if abbreviations are required  by the journal but if required, can this be presented in alphabetic order?

References:

A few reference citations are missing author(s) or have incomplete citations (volume and page numbers)

Reviewer 2 Report

The manuscript investigates the potential beneficial effect of xylo-oligosaccharides supplementation on bone properties, morphology of intestine, cecum parameters and related calcium transporters in growing mice model.

While the study was well-conducted, there are some concerns:

  • title -I suggest to remove the information how the XOS was administered and new title: “Effect of xylo-oligosaccharides supplementation on bone properties and related calcium transporters in growing mice” without dot at the end
  • in general, in the introduction Authors cite the review literature too often - I suggest citing original research papers
  • line 36: “At present,  the  public  prefers  to  pay  attention  to  some  functional ingredients that improve Ca absorption” – this reference is not the best one because it concerns the absorption of magnesium - I suggest you find the literature on calcium absorption, e.g. Nutrients 2017, 9, 702; doi:10.3390/nu9070702
  • line 43: instead “flora” please write “microbiota”
  • line 56: “L. paracasei” write using italics
  • line 57 and 79: explain the abbreviation “HFD“; “ICR”
  • line 65: instead “chance” write “change”
  • line 73: “expression of several molecules of calcium ion…” – quite unfortunate – please rewrite
  • line 104: samples were collected from orbital vein before sacrifice – could you confirm that it was “orbital” vain
  • line 119: provide the details of DEXA producer
  • page 12, line 42: again instead “flora” please write “microbiota”
  • page 12, line 42: put the reference number directly after Darío Pérez-Conesa et al [35]
  • page 12, line 74-77: “…some clinical trials have pointed out that the effect of NDOs …."– clinical trials by definition are are a type of research that studies new tests and treatments and evaluates their effects on human health outcomes. None of the references 40-43 are clinical trials, besides there references are quite old. In addition, research on the animal model serves as an attempt to explain the phenomena and mechanisms occurring in the human body, but not the other way around - please rewrite this sentence
  • Most of the results are presented in tables, but there are too many of them - I suggest you combine some tables into one common or try to present some results in graphic "user-friendly" format
